# Continuous and low-carbon production of biomass flash graphene

Xiangdong Zhu [1,2,10] ✉, Litao Lin[1,3,10], Mingyue Pang[4,10], Chao Jia[1], Longlong Xia[2], Guosheng Shi[5], Shicheng Zhang[1], Yuanda Lu[1], Liming Sun[1], Fengbo Yu[1], Jie Gao[1], Zhelin He[1], Xuan Wu[1], Aodi Li[1], Liang Wang[3], Meiling Wang[6], Kai Cao[6], Weiguo Fu[6], Huakui Chen[6], Gang Li[7], Jiabao Zhang[2], Yujun Wang[2] ✉, Yi Yang[4] ✉ & Yong-Guan Zhu[8,9] ✉

Flash Joule heating (FJH) is an emerging and profitable technology for converting inexhaustible biomass into flash graphene (FG). However, it is challenging to produce biomass FG continuously due to the lack of an integrated device. Furthermore, the high-carbon footprint induced by both excessive energy allocation for massive pyrolytic volatiles release and carbon black utilization in alternating current-FJH (AC-FJH) reaction exacerbates this challenge. Here, we create an integrated automatic system with energy requirement-oriented allocation to achieve continuous biomass FG production with a much lower carbon footprint. The programmable logic controller flexibly coordinated the FJH modular components to realize the turnover of biomass FG production. Furthermore, we propose pyrolysis-FJH nexus to achieve biomass FG production. Initially, we utilize pyrolysis to release biomass pyrolytic volatiles, and subsequently carry out the FJH reaction to focus on optimizing the FG structure. Importantly, biochar with appropriate resistance is self-sufficient to initiate the FJH reaction. Accordingly, the medium-temperature biochar-based FG production without carbon black utilization exhibited low carbon emission (1.9 g $CO_2$-eq g$^{-1}$ graphene), equivalent to a reduction of up to ~86.1% compared to biomass-based FG production. Undoubtedly, this integrated automatic system assisted by pyrolysis-FJH nexus can facilitate biomass FG into a broad spectrum of applications.

Flash Joule heating (FJH) is a burgeoning technology for upcycling biomass to few-layered flash graphene (FG)[1–7]. However, traditional pyrolysis by thermal radiation can only produce graphite-like material from biomass due to the lack of exfoliation process[8–10]. With great potential in diverse graphene applications, it is time to bring biomass FG out of the laboratory and into limelight[11–13]. However, developing continuous fabrication devices to implement scale production of biomass FG is a prerequisite for its successful implementation[14–17].

In the past three years, FJH technology has been developed in two generations succeeding in producing FG from carbonaceous waste[1,5,18–20]. First-generation FJH fabrication technology is in manual mode and delineating blueprints for continuous production[1,21–23]. The FJH reaction is ultrafast (at the seconds scale), and the complete FG production rate was determined by the time-consuming manual loading, pumping, and unloading[24,25]. Fortunately, an automatic FJH system was just reported and improved FG production rate[4]. However, this improvement was at the expense of the negative pressure pumping step, leading to an oxidation reaction under high temperature of FJH reaction and subsequent yield decrease. Without doubt, an integrated automatic device is urgently needed to tackle the

bottlenecks of continuous biomass FG production. In addition, the carbon abatement issue via technical improvement must also be addressed[26,27]. Energy-excessive AC-FJH produces massive energy waste during biomass FG production. AC-FJH includes the carbonization and graphitization processes, but the ultrahigh temperature (~2000 K) required for carbonization is excessive[28–30]. Meanwhile, a recent life cycle assessment (LCA) indicated that pyrolytic volatiles release of biomass in AC-FJH step is the main driver of the carbon emissions on FG production, accounting for 61.7%–77.7%[5]. Thus, this energy allocation is unreasonable, and we should allocate more energy on FG structure optimization. In addition, adequate carbon black must be added as a conductive agent to initiate the FJH reaction owing to the high resistance of biomass[31–33]. However, carbon black production is energy-intensive and contributed 5.89%–10.7% to the carbon emissions of FG production[5,34]. Therefore, pyrolytic volatiles release and carbon black utilization in the AC-FJH step are the leading contributor to the carbon emissions on FG production. Undoubtedly, reducing excessive energy on pyrolytic volatiles release and carbon black utilization play a key role in carbon abatement. To address these challenges, we built an integrated automatic device and proposed a pyrolysis-FJH nexus to achieve continuous biomass FG production with low carbon emissions. In FG production process, the biomass pyrolytic volatiles are initially released through pyrolysis, followed by the implementation of FJH reaction to specifically focus on optimizing the FG structure.

## Results and Discussion
### Design of integrated automatic device
The integrated device was controlled by a programmable logic controller via assembling modular components (including mechanical control, FJH reaction, and electrical control) (Fig. 1a, Supplementary Figs. 1–6, and Table 1, 2). Instead of performing manual loading and unloading, robotic arms were constructed to achieve continuous biomass FG production[35,36]. However, a small cumulative deviation may exist between the coordinate systems of the robot arms and the sample tray, resulting in samples not being successfully grabbed. To solve this problem, we employed an imputation algorithm based on locking the coordinates of the first and last samples onto a sample tray. Moreover, we proposed a movable sleeve to open and close the reaction zone for a negative-pressure environment. The construction of the negative-pressure environment was based on the time it takes to reach the negative pressure by pumping.

The critical flaw in the mechanical design was overcome, followed by the design of the electrical system. We first used the sequential function chart (SFC) program language to divide the biomass FG fabrication process into programmable discrete function blocks (such as the FJH reaction), and then cascaded the function blocks into a structured program based on the logical order of the production processes (Supplementary Fig. 7). Because the biomass FG fabrication process prioritizes AC-FJH followed by direct current FJH (DC-FJH), the introduction of AC into the integrated device may lead to the breakdown of DC discharge capacitors. Therefore, we realized the cyclic discharge of AC and DC using an AC contactor to cut off the DC discharge circuit during AC-FJH. Based on the above mechanical and electrical design, the production rate per batch can reach four times higher than that of first-generation biomass FG fabrication technology (Fig. 1b). And it is higher than the first and second-generation fabrication technology. A sample tray of 16 samples was produced in ~8 min with a high yield of 21.6 g h$^{-1}$ owing to the high device stability (Fig. 1c and Supplementary Movies 1, 2). We can further increase the single-batch biomass FG yield by augmenting the capacitors.

### Structure verification of flash graphene
As mentioned previously, the massive unrestrained release of biomass pyrolytic volatiles (H$_2$ account for 21.7–24.5 vol %, Supplementary

Table 3) during the AC-FJH process can contribute to a high carbon footprint (Fig. 2a, b, Supplementary Figs. 8–13). Therefore, we proposed an FJH with pyrolysis pre-treatment strategy to achieve FG production with low carbon emissions. Owing to the high resistance of the biomass and low-temperature biochar (300 and 600 °C), carbon black was required to reduce the sample resistance to initiate the AC-FJH reaction (Supplementary Fig. 14). However, the resistance of medium- (750 °C) and high-temperature (900 °C) biochar is appropriate to initiate the AC-FJH reaction without carbon black. Optimized production scenario with few-layered FG structure is a prerequisite for carbon emissions accounting. Therefore, the structures of biomass FG from various paths were evaluated first. The Raman spectra indicated that FG from biomass, few-layered graphene derived from low- and medium-temperature biochar could be produced at relatively low DC discharge voltage (Fig. 2c). In contrast, the fabrication of high-temperature biochar-based few-layered graphene required a higher DC discharge voltage (Supplementary Fig. 15). More information about the biomass FG structure is provided in the Supplementary Figs. 16–18[37].

As shown in the equivalent circuit diagram of DC-FJH (Fig. 2d), the biomass FG structure can be adjusted by increasing the sample-to-device resistance ratio to increase the sample-allocated voltage (Supplementary Table 4). It was noted that the few-layer structure could be achieved from low- and medium-temperature biochar-based preliminary FG due to appropriate sample-allocated voltage (81.2–123 V), at the sample-to-device resistance ratio ranging from 3.2 to 17.8. However, the low sample-to-device resistance ratio (2.4) of high-temperature biochar-based preliminary FG led to insufficient voltage allocation at relatively low DC discharge voltage, resulting in inadequate graphitization (Supplementary Fig. 19a, b). Therefore, the sample-allocated energy could be increased by increasing the voltage to achieve few-layered biomass FG production derived from high-temperature biochar. Meanwhile, a similar trend was also observed in the medium-temperature biochar-based FG production process (Supplementary Fig. 19c, e, Supplementary Figs. 20, 21, and Tables 5, 6). These results demonstrated that confined energy partition induced accurate graphitization results in a successful synthesis of few-layer graphene from five production paths.

### Carbon accounting for flash graphene
It should be noted that the required energy for FG structure formation is similar for all five pathways (Supplementary Fig. 22a). However, the contribution of structural optimization to carbon emissions accounts for only 9.74% in biomass-involved FG production pathway (Fig. 3a). Massive energy was wasted on biomass pyrolytic volatiles release rather than FG structure optimization (Supplementary Fig. 22b). Obviously, carbonization can be performed at lower temperatures and does not require an energy-intensive AC-FJH process. Therefore, the biomass-involved FG production pathway required more AC-FJH reaction batches due to unreasonable energy allocation (Fig. 3b). The high loss1 value in Path A is caused by the high reaction batches (Supplementary Table 7). In the biochar-involved FG production path, we used low-energy pyrolysis to prioritize removing the volatiles released in the energy-intensive AC-FJH process, while AC-FJH was only used for further graphitization. Therefore, low-carbon biomass FG production can be achieved through appropriate energy allocation, which is called energy cascade requirement (Fig. 3c). Accordingly, Fig. 3d shows that the carbon emissions of biochar-involved FG production paths were significantly reduced by 80.1–86.1% compared to biomass-involved FG production (Supplementary Table 8). Undoubtedly, various energy allocations in biomass FG production can reach similar destinations, as confirmed by material flow analysis of the biomass-to-FG path (Supplementary Fig. 22c, d). In addition, the reduction of both carbon black and quartz tube (by reduction of AC-FJH batches) utilization further

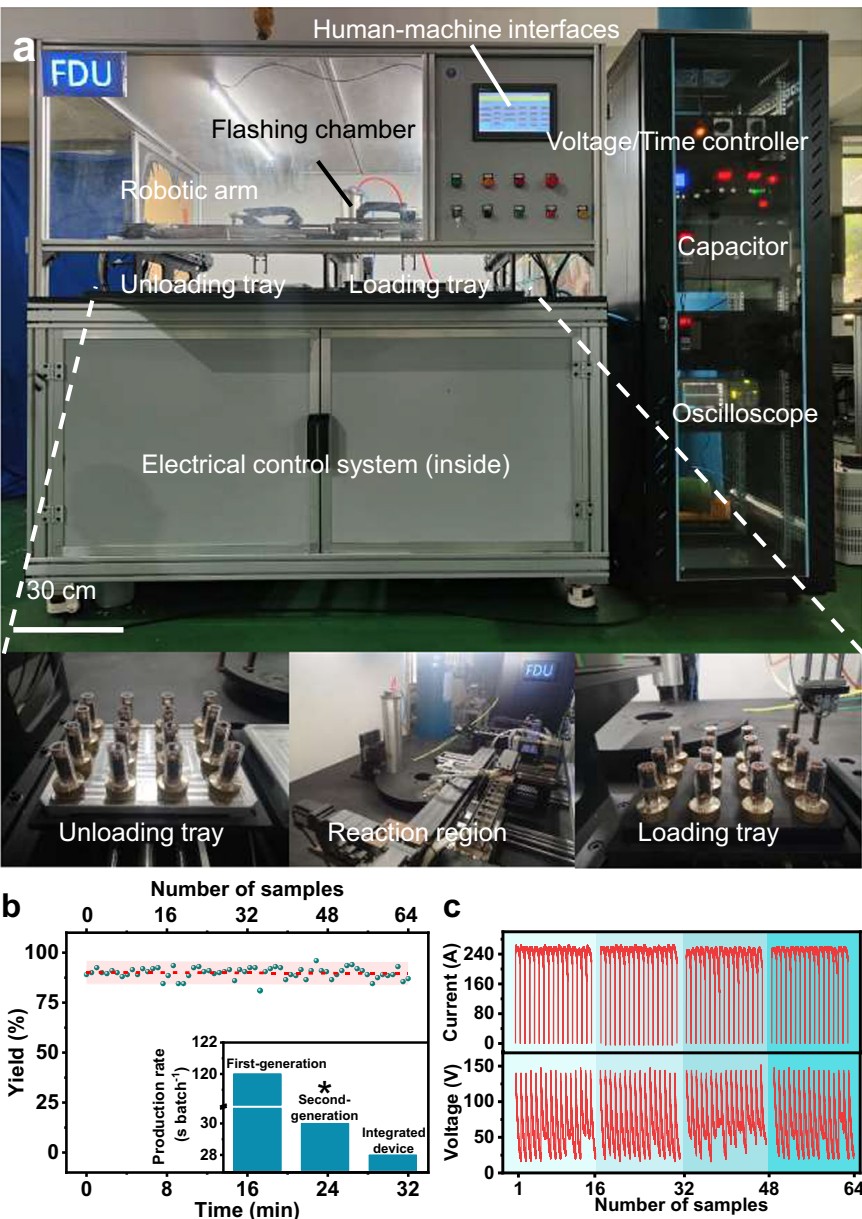

**Fig. 1 | Design and creation of continuous flash graphene production device.**
**a** Photograph of pilot-scale fabrication device of flash graphene. **b** Measured yield of 64 biomass flash graphene in 4 batches. Red dotted line is a linear fitted curve fitted to the sample yield. The pink-highlighted area is the fitted 95% prediction band according to the linear fitted curve. Insert production rate per batch of first-, second-, and third-generation (integrated device, this work) fabrication technology. First-generation fabrication technology means the biomass flash graphene by

FJH at lab-scale (refs. 1,5). Asterisk note: the second-generation fabrication technology (ref. 4) exhibits a high production rate at the expense of the pumping step. The lack of a pumping device will lead to a flame during the FJH reaction process, affecting the biomass flash graphene yield and accompanying safety risks. **c** Continuous current and voltage records of 64 biomass flash graphene production during direct current flash Joule heating by oscilloscope.

contributed to a lower carbon emission (Fig. 3a). Moreover, the 750 °C biochar-involved FG production process has a slightly higher economic benefit than other biochar-involved production paths owing to the absence of carbon black and suitable pyrolysis temperature (Supplementary Fig. 22e). Such a profitable production path with excellent graphene structure could be the best choice for continuous production at the pilot scale. Furthermore, the biomass FG systems from the pyrolysis-FJH nexus had much smaller impacts than the biomass-involved production processes across all impact categories evaluated (Fig. 3e). This is mainly because biochar-involved production process consumes the least electricity during the AC-FJH and DC-FJH processes, approximately half of that consumed by the biomass-based FG system.

## Continuous production of flash graphene

We optimized the FJH reaction by regulating the discharge voltage and sample loading weight at the pilot scale to overcome the amplification effect between the lab- and pilot-scale devices (Supplementary Fig. 23). The sample per batch has a few-layer structure owing to the steady continuous production process, which was also confirmed by Raman, XPS, AFM, and TEM analyses (Fig. 4a, b and Supplementary Figs. 24–34, Table 9). The carbon content of the sawdust FG was up to ~97.3%, indicating that the as-synthesized biomass FG had a high purity due to the low impurity content in the parent biomass (Supplementary Figs. 35, 36 and Tables 10 and 11). Sawdust FG exhibited excellent dispersibility, catalytic (bromate removal, ~93.3%) and solar absorption (~92.4%) performance, which are comparable to those of common

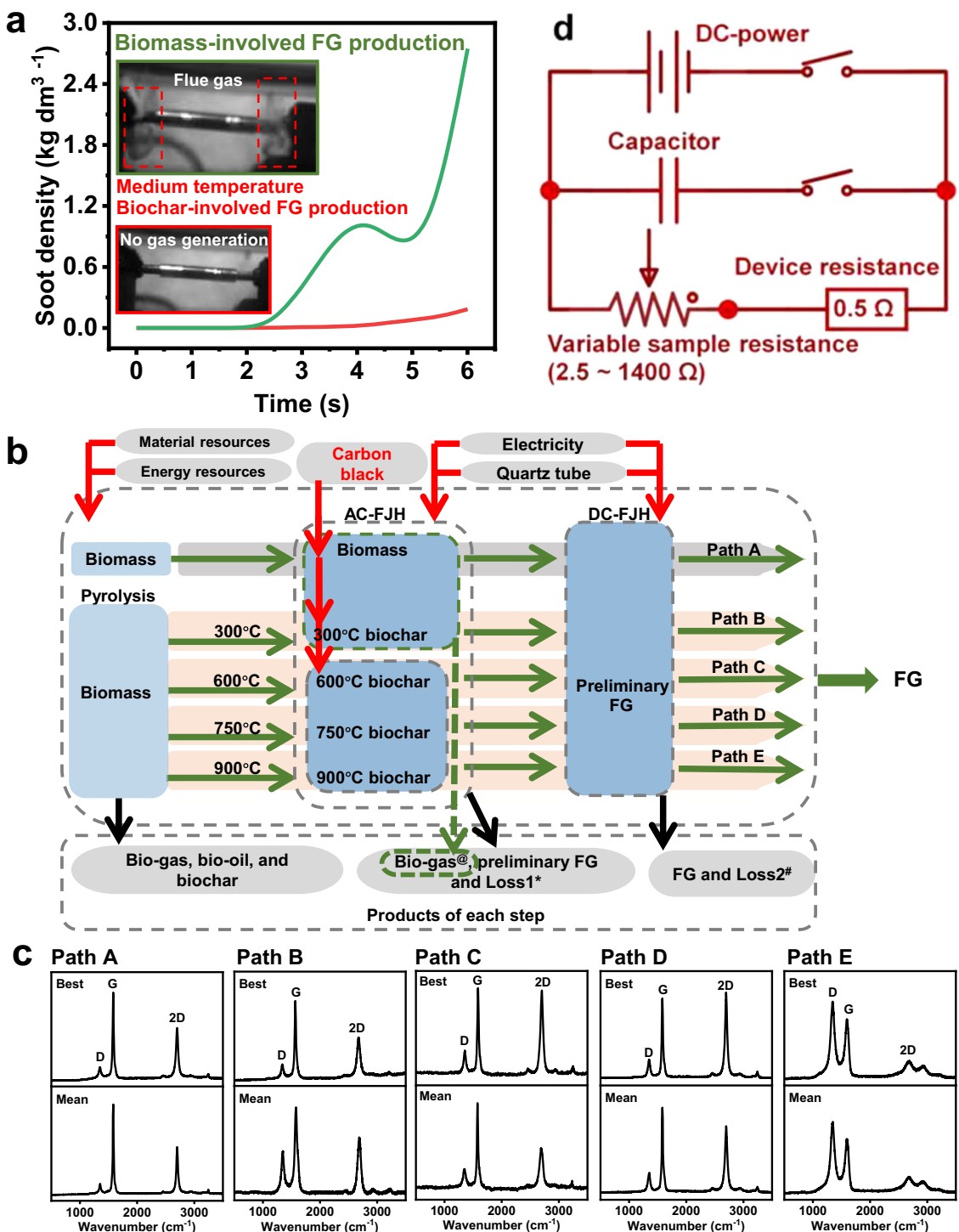

catalytic and photothermal materials[38–40] (Supplementary Figs. 37–39). Meanwhile, the sawdust FG production process had low carbon emissions at the pilot-scale (1.91 g $CO_2$-eq $g^{-1}$ graphene), similar to the lab-scale sawdust FG production. Therefore, employing an integrated device with a carbon emissions contribution of 26.6% does not lead to an apparent increase in carbon emissions.

In addition to sawdust, we examined the FG production using another representative low-impurity biomass, bamboo, to further demonstrate that the advantage of the system in producing high-purity graphene. Owing to the few-layer structure, the excellent dispersibility and catalytic properties of bamboo FG were comparable to those of the fabricated sawdust FG (Supplementary Fig. 40a–f,

Supplementary Figs. 41–43, and Supplementary Table 12). Moreover, the pyrolysis-coupled bamboo FG production process had low-carbon emissions (1.90 g $CO_2$-eq $g^{-1}$ graphene), which is comparable to sawdust FG production (Supplementary Fig. 40g, h).

We used rice straw as a precursor to demonstrate that FG could be fabricated from biomass with high impurities (Supplementary Fig. 44a–c). Rice straw-derived FG exhibited a few-layer structure but had a low yield owing to metal impurity volatilization during FJH reaction, which induced numerous carbon element losses (-51.3%, Supplementary Fig. 44d, and Supplementary Fig. 45, Tables 13, 14). Biomass FG derived from rice straw had a high impurity content (~33.2%) owing to the residual impurity in the FJH reaction process, as

**Fig. 2 | Fabrication path and process of biomass/biochar-based flash graphene.**
**a** Soot density of biomass/biochar-involved flash graphene production during alternating current flash Joule heating was simulated by fire dynamics simulator. Inset: digital images of biomass-involved flash graphene production during alternating current flash Joule heating with volatiles emission (upper) and 750 °C biochar-involved flash graphene production during alternating current flash Joule heating without volatiles emission (below). **b** Life cycle assessments system boundary of biomass/biochar-involved flash graphene production process. The produced bio-gas (such as $H_2$ and CO) and bio-oil can be collected for fossil fuel substitution. Path A: Biomass is first treated by AC-FJH to produce preliminary FG, and then preliminary FG is fabricated by DC-FJH to produce FG; Path B: Biomass is first pyrolyzed to release pyrolytic volatiles and produce 300 °C biochar, then biochar treated by AC-FJH to produce preliminary FG, and finally preliminary FG is fabricated by DC-FJH to produce FG; Path C: Biomass is first pyrolyzed to release

pyrolytic volatiles and produce 600 °C biochar, then biochar treated by AC-FJH to produce preliminary FG, and finally preliminary FG is fabricated by DC-FJH to produce FG; Path D: Biomass is first pyrolyzed to release pyrolytic volatiles and produce 750 °C biochar, then biochar treated by AC-FJH to produce preliminary FG, and finally preliminary FG is fabricated by DC-FJH to produce FG; Path E: Biomass is first pyrolyzed to release pyrolytic volatiles and produce 900 °C biochar, then biochar treated by AC-FJH to produce preliminary FG, and finally preliminary FG is fabricated by DC-FJH to produce FG. *Note: "Loss1" refers to the bio-oil and depletion in AC-FJH. #Note: "Loss2" refers to the pyrolytic volatiles (probably bio-oil or gas) and depletion in DC-FJH. @Note: In the AC-FJH reaction, bio-gas is produced only in the path of biomass and 300 °C biochar as feedstock. **c** Intensity ratio of the 2D and G bands (in Raman spectra) of flash graphene fabricated from Paths A-E. **d** Equivalent circuit diagram of direct current flash Joule heating system.

confirmed by XRD (Supplementary Fig. 46). While a high impurity content did not affect the photothermal absorption capacity (~94.9%, Supplementary Fig. 47), the dispersion and catalytic properties of biomass FG from rice straw were lower than those of high-purity biomass FG (Supplementary Fig. 44e, f). Furthermore, the carbon emissions of rice straw FG from the pyrolysis-FJH nexus production path were slightly higher than those of high-purity FG (Supplementary Fig. 44g, h). This is because low purity FG production with a lower yield than high purity FG production requires more AC-FJH reaction batches, resulting in high electricity consumption.

In summary, the biomass FG production path with pyrolysis pretreatment in this study has lower carbon emissions than previously reported biomass-based FG (Fig. 4d). Notably, if the electricity system is decarbonized through the use of renewable energy, then biomass FG systems could even achieve carbon-negative values considering the fossil fuel offset scheme (Fig. 4d). In terms of carbon emissions and financial benefits, the pyrolysis-FJH nexus production technology of high-purity FG outperforms the traditional graphene production technology (Fig. 4e and Supplementary Tables 15–18). Inexhaustible low-impurity biomass (such as sawdust), with more than 100 million tons of global production[41], can satisfy high-purity graphene production on a million-ton scale via the integrated system with an amplified capacitor in the future (Fig. 4f). Such high-purity biomass FG has few-layer structure, and can be used for thermal conductivity, photothermal, catalysis, and cement fillers (Supplementary Table 19). Meanwhile, biomass with high impurities (such as rice straw) could affect the purity and yield of biomass FG but does not affect its application as a photothermal material. Therefore, owing to the breadth of raw materials, solvent-free addition, and low energy consumption, continuous FG production technology could provide an option for few-layered graphene production.

## Methods

### Integrated automatic device for continuous flash graphene production

Integrated and automatic device was designed to progress the continuous production of the FG. The design philosophy of this device is to ensure the integrity of each FG production process, including loading, negative-pressure pumping step, FJH reaction, and unloading. Therefore, we integrated the multiple modular components for complete production turnover, including mechanical control, FJH reaction, and electrical control areas (Supplementary Fig. 6 and Supplementary Table 1, 2). The main objective of the mechanical control area design is to replace the manual operational parts (loading and unloading) with robot arms. For the FJH reaction area design, we simultaneously constructed a negative-pressure environment and amplified the reaction zone by a movable sleeve with pumping function and a larger reaction electrode. The pumping duration to reach the negative-pressure environment was recorded and inputted into the integrated device.

In the electrical control area, the FG fabrication process was divided into programmable action and transfer conditions by SFC program language for automation. SFC is a programming language for satisfying sequential logic control. The sequential logic control system is assigned according to the transfer conditions, and the action is carried out step by step in accordance with the FG production process. Furthermore, we use the imputation algorithm to solve the grab error of the robot arm, which is one of the technological bottlenecks in the electrical control design. The imputation algorithm is used to calculate the coordinates of each sample. Thus, this algorithm eliminates the error caused by mechanical installation between the sample tray and the robot arms.

On the human-machine interfaces, pumping time, motor running speed, and FJH reaction voltage/time can be regulated (Supplementary Figs. 1, 2). Before continuous production, the robotic arm, sleeve, and servo motor should be initialized by clicking the reset button (Supplementary Fig. 3). Then, observing that the "startup ready" light is lit, and the system enters the ready state for automatic operation. If the "startup ready" is off, perform the above initialization steps again. After system initialization, press the "start" button to make the system operate automatically. Generally, the continuous automatic production process is as follows (Supplementary Movies 1, 2): 750 °C biochar was first loaded in a quartz tube and pressed tightly with graphite electrodes. Subsequently, the quartz tube was vertically placed on the sample tray and then grabbed by the robot arm onto the cam. Then, the sample is rotated to the reaction zone, the two-terminal copper electrodes clamp and compress tightly for sufficient FJH reaction. Finally, the robot arms grab the as-synthesized FG to the blanking zone after the FJH reaction. After one cycle FG fabrication process, the following procedures are cycled to synthesize the FG continuously. During continuous production, click the "signal monitoring" option on the homepage to enter the monitoring page to check the operational status (Supplementary Figs. 4–6 and Supplementary Table 1). Precautions and instructions for programmable integrated device are provided in Supplementary note: precautions and instructions of integrated device.

To obtain continuous biomass FG production, effect of the sample loading weight on the FG layer was investigated (0.1, 0.2, and 0.4 g). A total of 750 °C biochar-based FG from various biomass (including sawdust, bamboo, and rice straw) was wrapped with graphite electrodes in quartz tubes (tube thickness: 3 mm, inner diameter: 10 mm, length: 35 mm) and placed on the loading sample tray. The continuous production process is described above. Furthermore, the effect of the DC reaction voltage (200 and 250 V) on FG layer was also examined.

### Flash graphene synthesis for carbon emission accounting

The optimized production scenario with few-layered FG structure is a prerequisite for carbon emissions accounting. Therefore, the structures of biomass FG from various paths were first evaluated. Various

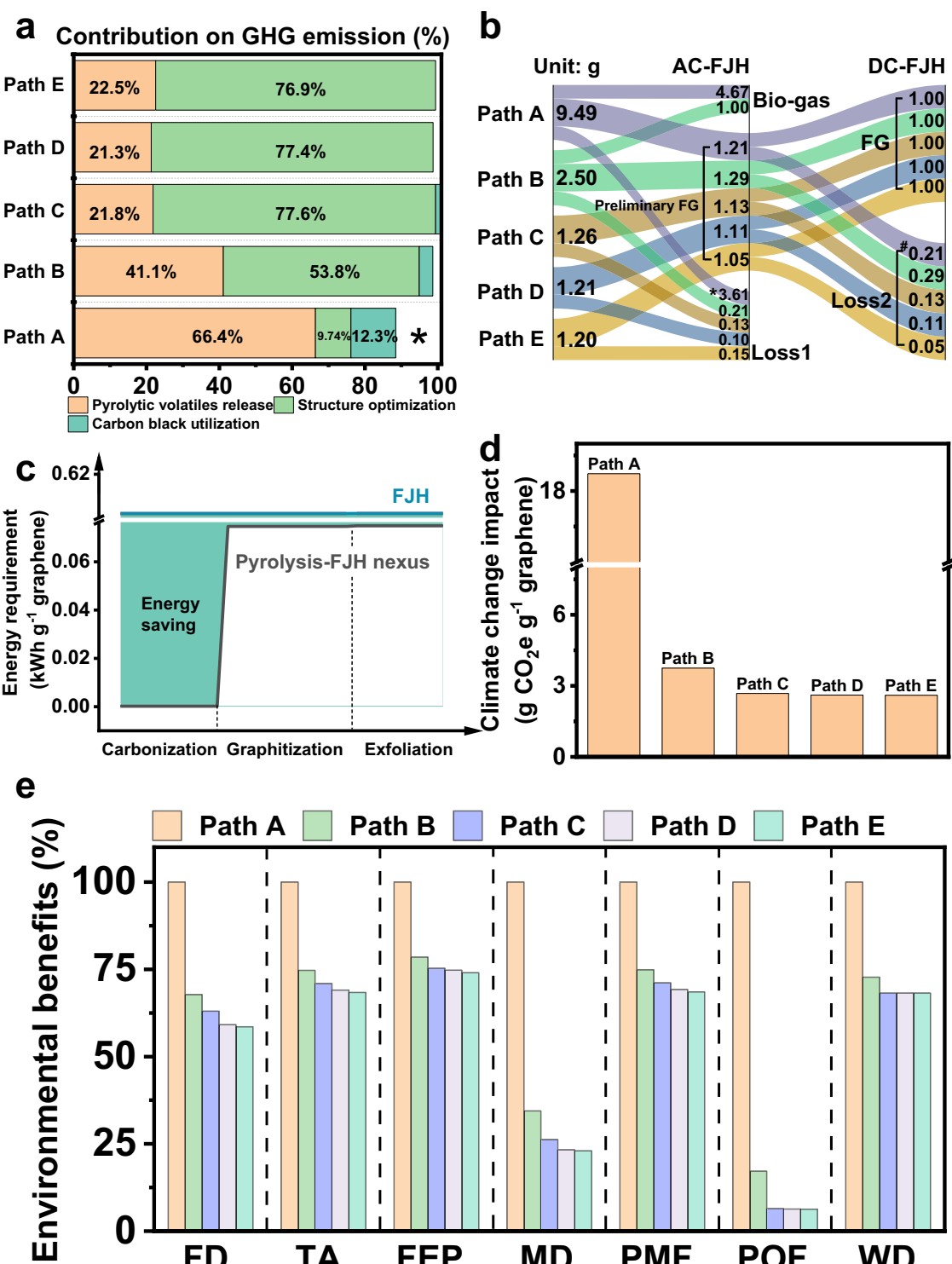

biomass (including sawdust, bamboo, and rice straw) was employed for AC-FJH and DC-FJH. Briefly, the biomass was crushed into powder (80 mesh) and carbonized to biochar (300, 600, 750, and 900 °C). Then, 0.1 g of the biochar (>700 °C) was loaded in a quartz tube (tube thickness: 2 mm, inner diameter: 6 mm, length: 45 mm) to initiate FJH reaction. Subsequently, the sample was placed in a vacuum desiccator (~0.6 psi) and operated on alternating current (200 V, 50 Hz) for ~6 s to produce preliminary FG. Subsequently, 0.1 g of preliminary FG was subjected to DC-FJH to obtain few-layered FG at the desired discharge voltage and time. Moreover, the resistance of biomass and biochar (<700 °C) is too high to initiate the FJH reaction. Thus, the FG

fabrication from these high-resistance parent materials requires adding 5% carbon black as conductive agent. The end of each copper electrode was hollow to facilitate the collection of the generated bio-gas with a gas bag (200 mL). In the AC-FJH process, high temperature (~2000 K) and high current (~25 A) can be generated. Non-condensable bio-gas (including $H_2$, $CH_4$, CO, $CO_2$, $C_2H_2$, $C_2H_4$, $C_2H_6$, $C_3H_6$, $C_3H_8$) were quantitatively determined from AC-FJH reaction of biomass and 300 °C biochar, respectively (Supplementary Table 3). This is because biomass and 300 °C biochar as carbon precursors have a low aromatization, as proved by the high H/C. However, the bio-oil can be condensed on quartz tubes due to the ultrafast cooling rate. And, this part

**Fig. 3 | Life cycle assessment of various flash graphene (for one gram graphene production). a** Contribution analysis on the GHG emissions derived from 1 g of flash graphene produced by the five production systems in terms of pyrolytic volatiles release, structure optimization, and carbon black utilization. Asterisk note: for the FG system of path A, quartz tubes phase contributes 11.1% to the total climate change impact. Quartz tubes consumed in Paths B-E make negligible contributions (less than 1%). **b** Material flow of five different biomass flash graphene production paths from biomass (Path A) and biochar (Path B-E) to flash graphene. "Loss1" refers to the pyrolytic volatiles (bio-oil as the main composition) and depletion in AC-FJH. "Loss2" refers to the pyrolytic volatiles (probably bio-oil or gas) and depletion in DC-FJH. *Note: for 1 gram graphene production, 92 times AC-FJH reactions are required in Path A, while only 11 times in Path C-E. Therefore, a high accumulated depletion value is formed in Path A. Overall, a high loss value in path A is formed. #Note: for 1 gram of graphene produced, 11- and 12-times DC-FJH

reactions are required in Path A and B, while only 6 times in the Path C-E. Therefore, a high accumulated depletion and loss value in paths A and B is formed. **c** Comparison of the energy requirement of biomass-based flash graphene production path by FJH with biochar-based flash graphene by pyrolysis-FJH nexus. The green-highlighted area represents the energy gap between the two paths. The flash graphene fabrication process is divided into carbonization, graphitization, and exfoliation. **d** Comparison of life cycle GHG emissions between biomass-based flash graphene system and biochar-based flash graphene systems. **e** Life cycle environmental impacts of biomass-based flash graphene system (Path A) and biochar-based FG systems (Path B-E). For each impact category, the highest impact value is set to be 100%, and the value of other paths equals the percentage shares of each product system based on this highest impact. FD fossil depletion, TA terrestrial acidification, FEP freshwater eutrophication, MD metal depletion, PMF particulate matter formation, POF photochemical oxidant formation, WD water depletion.

is difficult to collect and quantitatively analyze. In addition, the sample depletion was inevitably caused by high-temperature induced ejection and adhesion of the sample on the copper wire mesh and quartz tube. Therefore, we named "Loss 1" for the bio-oil and depletion. Therefore, we named "Loss 1" for the bio-oil and depletion during AC-FJH.

In DC-FJH process, the ultrahigh temperature and current (~3000 K, ~150 A) can be generated. Such an ultrahigh temperature far exceeds the previous AC-FJH reaction (~2000 K) and will further prompt graphitization degree of preliminary FG, as confirmed by the increased H/C (Supplementary Tables 5, 6). This process will inevitably cause the release of pyrolytic volatiles (possibly non-condensable bio-gas and condensable bio-oil), as confirmed by the increased carbon content (Supplementary Table 6). However, the amount of bio-oil and bio-gas produced is too small to be collected for qualitative and quantitative analysis. In addition, the sample depletion was inevitably caused by high current (~150 A) induced ejection of the sample on the copper wire mesh. Therefore, we named "Loss 2" for the pyrolytic volatiles and depletion during DC-FJH.

## Structural analysis of biomass flash graphene
The layer and defect of FG were analyzed by Raman spectra using an XploRA Raman spectrometer with a laser wavelength of 532 nm, laser power of 5 mW and 50 × lens. In Raman spectra, three distinctive Lorentz peaks were fitted at D (~1350 cm$^{-1}$), G (~1580 cm$^{-1}$), and 2D (~2700 cm$^{-1}$) bands using Lab-Spec6.4 software. FG derived from various parent materials was subjected to X-ray diffraction (XRD) analysis using Rigaku Ultima IV with Cu Kα radiation (λ = 1.54 Å) in the 2θ range of 5–90° at a scanning rete of 10° min$^{-1}$. X-ray photoelectron spectroscopy (XPS) was performed on FG using Thermo ESCALAB 250 XI with Al Kα X-ray radiation (400 μm spot size). Survey scans were acquired with a pass energy of 50 eV and a step size of 0.05 eV. The binding energies of high resolutions spectra were calibrated using the C 1 s peak at 284.8 eV. To investigate the thermostability of feedstocks, TGA was conducted on Netzsch TG 209 F3 Tarsus at 800 °C (heating rate of 10 °C min$^{-1}$) for a duration of 60 min under an air flow rate of 100 mL min$^{-1}$. The elemental analysis (C/H/N/S) of FG was performed using an elemental analyzer (Vario EL III, Germany). The morphologies of FG were observed using transmission electron microscopy (TEM, Tecnai G2 F20 S-Twin, FEI). The flake thickness of FG was determined using an atomic force microscope (AFM, Asylum Research MFP-3D, USA). To investigate the sample-allocated-voltage controlled by DC discharge voltage on biomass FG layer, sample-allocated-voltage (V$_1$) is calculated by the following equation: $V_1 = R_1 / (R_1 + R_2) * V$, where the resistance of sample and device are R$_1$ and R$_2$ (0.5 Ω), and V is recorded voltage by oscilloscope. To simulate and analyze the transient motion of flue gas from biomass during AC-FJH, we used Fire Dynamics Simulator (FDS) software (Version 6.3.2). Considering that the software has the characteristic of rectangular modeling, the cuboid computing space was established with 24 cm × 24 cm × 26 cm. Based on ignoring the pore effect and volatiles

condensation, we adopt a finite rate reaction model including carbonization and Lagrange high-temperature ignition form. The soot density, visibility, and velocity of flue gas were numerically analyzed. In the simulation process, a rectangular grid with a mesh size of 1 cm was used with 14,976 grids.

## Life cycle assessments of flash graphene production
Life cycle assessments (LCA) in this work was to evaluate the environmental impacts of various FG production process from cradle to gate[5]. The system boundary of the FG production covers the three main phases: pyrolysis, AC-FJH, and DC-FJH processes. We combined experimental data, literature, and databases to estimate the environmental impacts of FG production. The utilization of inputs may result in diverse emissions, which were estimated through the application of emission factors and methodologies established in previous research studies[42]. The environmental impacts associated with sawdust and bamboo production were neglected in this study because it is residue from the wood processing industry and is usually considered waste that has zero emission burdens[43,44]. The data on crop cultivation, including the usage of pesticides, fertilizers, and diesel fuel, were collected at the provincial level from relevant literature and statistical yearbooks. Specifically, the environmental impact attributed to rice cultivation accounted for 13.9% of the total environmental impact. The environmental impacts associated with sawdust and bamboo production were not considered in this study as they are residues from the wood processing industry and are typically regarded as waste with negligible emission burdens. In terms of renewable energy for decarbonizing FG production, hydroelectricity was assumed to be utilized due to its contribution to China's current renewable electricity generation.

The life cycle inventory data for the three conventional graphene production technologies is sourced from Cossutta et al. without any modifications[45]. The life cycle burdens of energy, agrichemical inputs, and other materials (such as electricity, fertilizers, carbon black, and quartz tube) were obtained from the Ecoinvent 3.5 database and the PE International database using the Gabi 8 platform's Cut-off System Model[46]. During the life cycle impact assessment (LCIA) phase, inventory results are translated into environmental impact potentials to comprehend their relative environmental significance. The LCIA model utilized in our study is the ReCiPe 2016 method.

## Cost-benefit analysis of flash graphene at pilot-scale
This paper undertakes an economic evaluation of conventional graphene production methods (including chemical oxidation and reduction alongside FJH technology). For the cost-benefit analysis, we computed net economic gains associated with pilot-scale biochar-based FG production. Net economic benefits were determined by subtracting maintenance costs (e.g., initial capital plant expenses, electricity consumption, and staff wages) from the financial returns gained from selling graphene products and energy sources.

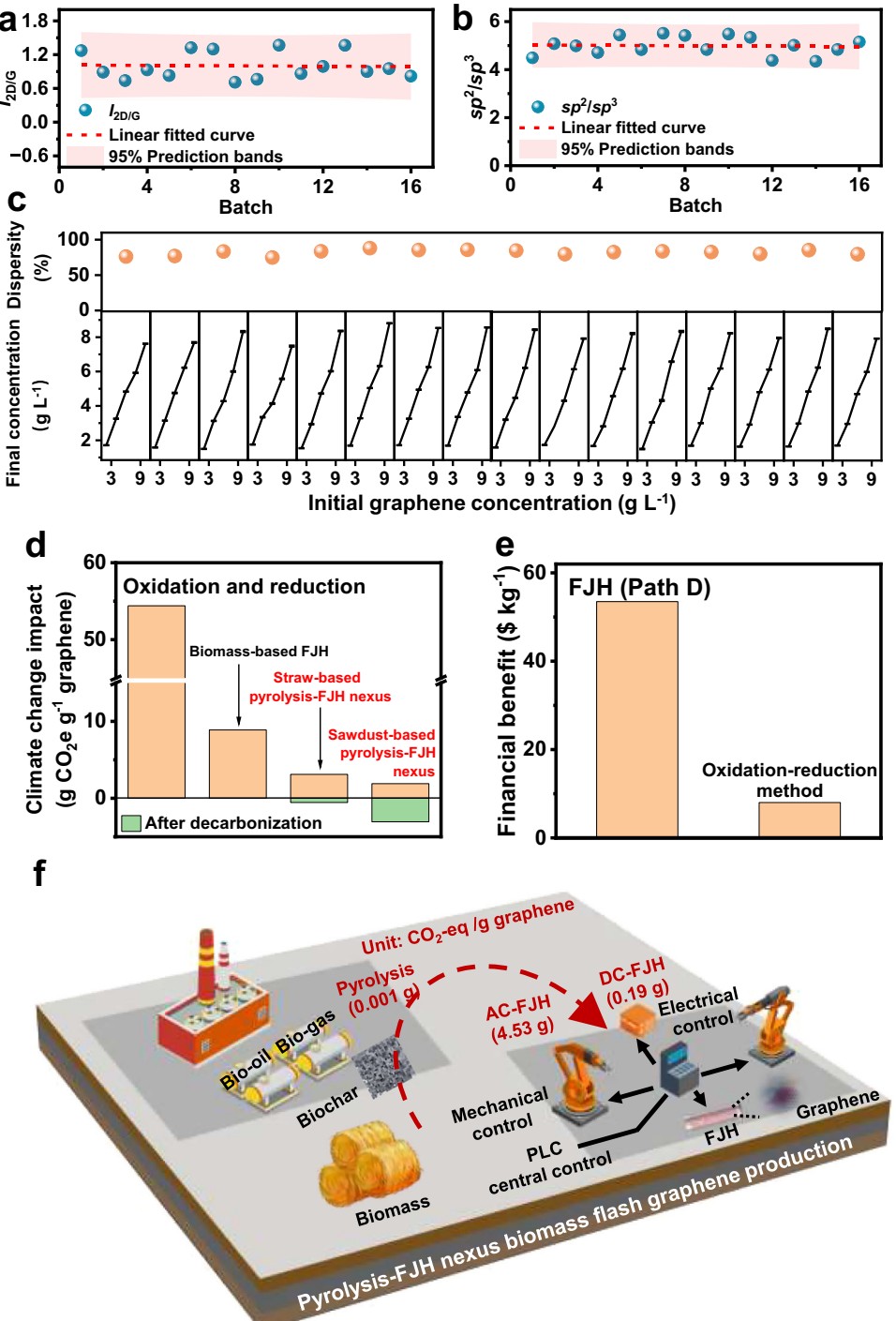

**Fig. 4 | Continuous fabrication of flash graphene. a** Intensity ratio of the 2D and G bands (in Raman spectra) of the biochar-based flash graphene at pilot-scale. We mixed every 4 samples into one batch to analyze the structure and application. To facilitate the analysis of the structure and application of the biochar-based flash graphene at pilot-scale, we mixed every 4 samples into one batch. **b** Intensity ratio of $sp^2$ and $sp^3$ carbon bond content of the biochar-based flash graphene at pilot-scale**. c** Upper: dispersity of flash graphene in a surfactant (F-127) solution. Below: flash graphene dispersion with various concentrations ranges from 2–10 mg L$^{-1}$.

The experiments were repeated twice. **d** Comparison of life cycle GHG emissions between biochar-based flash graphene systems at pilot-scale and industrial commodities (including aluminum, copper, steel and iron, petrochemicals, and cement). **e** Financial benefit comparison between biochar-based flash graphene system at pilot-scale and traditional mainly graphene production technology (oxidation-reduction). **f** Blueprint of the future industrial park for biomass flash graphene production.

GHG emissions resulting from the operation of FJH plants primarily stem from electricity consumption. Initial startup fuel is utilized to initiate the first pyrolysis unit, after which the process heat generated fuels subsequent units in a continuous cycle. In scenarios where significant quantities of graphene are produced through continuous

pyrolysis, the startup fuel's contribution becomes negligible and is often disregarded in techno-economic and LCA analyses.

Regarding material costs, dry waste biomass sourced from rural areas carries a market price of 28.2 US\$ Mg$^{-1}$. According to Wang et al.[47], the average transportation distance is 50 km, incurring a

transportation cost of 1.69 US\$ $t^{-1}$ $km^{-1}$. Industrial electricity prices stand at 0.08 US\$ $kWh^{-1}$. Labor cost, as outlined in Struhs's report (2020)[41], are incorporated within maintenance expenses. Three workers are assigned to the FG production department, while two workers are allocated to other sectors, including biochar production. Maintenance costs, derived from Struhs's report along with device costs[48], tally to an annual estimate of 31,000 US\$ $year^{-1}$. Repair service and maintenance fees are assumed to amount to 20% of the capital asset value[47,49]. Bioenergy predominantly consists of bio-gas and bio-oil. According to Azzi et al.[50], the conversion of bio-gas generated from the pyrolysis of 1 Mg of biomass into electricity is about 113.2 kWh. Market estimates peg the material price of graphene at 100 US\$ $kg^{-1}$ for graphene. As Wang's reported (2015)[47], a typical Salix direct-fired power generation system, per Wang's findings, was deemed economically unviable without government subsidies.

## Application evaluation of flash graphene

**Dispersion performance of biomass flash graphene.** FG was dispersed in water–Pluronic (F-127) solution (1%) at concentrations of 2–10 $g\,L^{-1}$. The mixture was sonicated in an ultrasonic bath for 40 min to obtain a dark dispersion. The dispersion was subjected to centrifugation at 251.6 × g (470 relative centrifugal force) for 30 min to remove aggregates using a Beckman Coulter Allegra X-12 centrifuge equipped with a 19-cm-radius rotor. The supernatant was analyzed via ultraviolet-visible spectroscopy (Shimadzu). The dispersions were diluted 300 times, and the absorbance was recorded at 269 nm.

**Biomass flash graphene for Pd-catalyzed hydrogenation.** To achieve highly efficient removal of bromate, Pd 1.5 wt% (from Sodium tetrachloropalladate (II), 99.99%) and flash graphene (1 $g\,L^{-1}$) were stirred together in a 100 mL round-bottom flask at room temperature under 1000 rpm for 10 min. Flash graphene could quickly trap $Pd^{2+}$ and enable it to disperse homogeneously. Then, the flowing $H_2$ was injected to activate the $Pd^{2+}$ reduction for 10 min. Subsequently, bromate solution (100 $mg\,L^{-1}$) was added for reduction during the dynamic process above by $Pd^0$ nanoparticles for 60 min. To observe the morphology of the flash graphene catalyst, transmission electron microscopy (TEM) was conducted. The chemical properties of the surface Pd on flash graphene was determined by X–ray photoelectron spectroscopy (XPS).

**Photothermal conversion performance of flash graphene.** Transmittance and reflectance of biochar-based FG were recorded to evaluate the solar absorption of the evaporator by using a UV-vis-NIR spectrometer (PE lambda 750) with an integrating sphere unit. The absorption of biochar-based FG was calculated by unit minus the reflectance and transmittance (Absorption=1-Reflectance-Transmittance).

**Thermal conductivity performance of biomass flash graphene.** Based on the transient planar heat source method, the thermal conductivity of biomass FG was measured by a thermal constant analyzer (Hotdisk 2500 S). The test method uses a helical heat source to measure the thermal parameters, referring to standard ISO22007-2. The sample is first pressed, and the probe is placed between two samples. The thermal conductivity of the biomass FG can be obtained by setting a constant heating power of the probe. By using the response of sample on the probe surface temperature, we are recording the probe temperature through the software and analyzing the data.

## Data availability

The data that supports the findings of the study are included in the main text, Supplementary Data 1, and other supplementary information files. Raw data can be obtained from the corresponding author upon request. Source data are provided with this paper.

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

## Acknowledgements

This work was supported by the National Natural Science Foundation of China (No. 22276040 to X.Z., No. 42225701 to Y.W.), the special fund of Yangtze delta region healthy agriculture institute, and leading project of Bai Ma future food research institute (JBGS-2021-007).

## Author contributions

X.D. Z. and L.T. L. conceived the research; M.Y.P. and Y.Y. conducted the life cycle assessment; Most experimental work (sample preparation, characterizations, and application) was performed by L.T. L. and assisted by C.J., L.L. X., G.S. S., S.C. Z., Y.D. L., L.M. S., F.B. Y., J. G., Z.L. H., X.W. and A.D. L.; L.W. developed the models and conducted the simulations; M.L. W., K. C., W.G. F. and H.K. C. built the integrated device; G. L., Y.J. W., J.B. Z. and Y.G. Z., compiled and discussed the results; X.D. Z. and L.T. L., wrote the manuscript. All authors reviewed the final manuscript.

## Competing interests

The authors declare no competing interests.

## Additional information

[1]Department of Environmental Science and Engineering, Fudan University, Shanghai 200433, China. [2]State Key Laboratory of Soil and Sustainable Agriculture, Institute of Soil Science, Chinese Academy of Sciences, Nanjing 210018, China. [3]School of Energy and Power, Jiangsu University of Science and Technology, Zhenjiang, Jiangsu 212003, China. [4]Key Laboratory of Three Gorges Reservoir Region's Eco-Environment, Ministry of Education, Chongqing University, Chongqing 400044, China. [5]Shanghai Applied Radiation Institute and State Key Laboratory Advanced Special Steel, Shanghai University, Shanghai 200444, China. [6]Institute of Intelligent Machines Hefei Institutes of Physical Science, Chinese Academy of Sciences, Changzhou 213164, China. [7]Key Lab of Urban Environment and Health, Institute of Urban Environment, Chinese Academy of Sciences, Xiamen 361021, China. [8]State Key Laboratory of Urban and Regional Ecology, Research Center for Eco-Environmental Sciences, Chinese Academy of Sciences, Beijing 100085, China. [9]Zhejiang Key Laboratory of Urban Environmental Processes and Pollution Control, CAS Haixi Industrial Technology Innovation Center in Beilun, Ningbo 315830, China. [10]These authors contributed equally: Xiangdong Zhu, Litao Lin, Mingyue Pang. ✉e-mail: zxdjewett@fudan.edu.cn; yjwang@issas.ac.cn; yi.yang@cqu.edu.cn; ygzhu@rcees.ac.cn

