## [Peer Review File · Nature Communications]

Continuous and low-carbon production of biomass flash grapheneEditorial Notes:

1. This manuscript has been previously reviewed at another journal that is not operating a transparent peer review scheme. This document only contains reviewer comments and rebuttal letters for versions considered at *Nature Communications*.
 2. Parts of this Peer Review File have been redacted as indicated to remove third-party material where no permission to publish could be obtained.
-

REVIEWER COMMENTS

Reviewer #1 (Remarks to the Author):

I am satisfied that the authors have responded to my comments and recommend this submission for publication, provided that the authors also include in the manuscript the information below that they included in their responses to me. The rebuttal letter should not merely be a communication with the review, but this needs to also be transmitted in the manuscript, please. After the small additions noted below, it will be ready. I do not need to review it again.

Q5. At the top of page 7, the authors state, “And it is higher than the first and second-generation fabricating technology.” Which fabrication technology are the authors referencing? They already mentioned flash graphene fabrication technology in the previous sentence.

The authors should also use the word “fabrication” instead of “fabricating”.

“Augmenting the capacitors” is also mentioned. What about the capacitors should be augmented? The number, the voltage, or the capacitance?

A: (1) First-generation fabrication technology means the biomass flash graphene by FJH at lab-scale (Refs. 1 and 5). The second-generation fabrication technology (Ref. 4) exhibits a high production rate at the expense of the pumping step¹. This has already explained in the legend in Figure 1.

(2) As your suggestion, we have changed the word “fabricating technology” to “fabrication technology”.

(3) “Augmenting the capacitors” refers to increase the number of series capacitors, thereby increasing the discharge voltage and ultimately improving the sample reaction mass of a single batch.

Please clarify in the manuscript that you are referring to the number of capacitors in series.

Q7. Figure 3. Why is exfoliation required for flash graphene production? Exfoliation is typically used for top-down methods, but not for bottom-up methods like flash Joule heating. The main text leads to extended data figures 5a-5b for more information on inadequate exfoliation, but the figures themselves do not discuss this.

A: We found that insufficient graphitization leads to incomplete graphene structure conversion. In extended data figures 5a-5b, the low sample-to-device resistance ratio (2.4) of high-temperature biochar-based preliminary FG led to insufficient voltage and temperature allocation at relatively low DC discharge voltage, resulting in inadequate graphitization.

Please include this explanation somewhere in the manuscript.

Q24. Supplementary table 5, 10, 11, 13, 14: how is H/C calculated? It does not appear to be H%/C% You should also be specific about whether your quantities are in wt% or atomic %.

A: The hydrogen-carbon ratio is calculated as follows: $H/C = (H \text{ relative weight} / H \text{ molecular weight}) / (C \text{ relative weight} / C \text{ molecular weight})$. H relative weight and C relative weight used in this equation are percentage by weight.

****Please include this information under the corresponding supplementary figures.****

Reviewer #2 (Remarks to the Author):

Comments on the paper NCOMMS-23-60727-T "Continuous and low-carbon production of biomass flash graphene" by Zhu et al.

The paper addresses an important Topic consisting in making graphenic-like materials from bioresources.

Specific comments:

Comment#1:

In Figure 2b, the biogas formation is claimed which is impossible because of the low Hydrogen content that is removed in the volatile at the drying and/ or pyrolysis step. The same comment is valid for Fig3a and 3b where the amount of volatiles is questionable. In fact, Bamboo used as biomass source is known to have very high silica content (inorganic composition not provided in this paper) that inhibit the pyrolysis and favor the biochar formation rather than the bio-oil or gas production. This is widely documented in the literature and is in contradiction with the author claims that are not substantiated. The authors failed in demonstrating an appropriate understanding of the pyrolysis of biomass.

Comment #2:

In Figure 3a: the pyrolytic volatiles rate seems not realistic so are the data on Figure 3b if we refer the composition of the initial feedstock and to the literature. Consequently, the data on Figure 3c are questionable too. For this figure, the authors could have substantiated the data and claims in providing the proximate and ultimate analysis and see the H/C and O/C ratio for instance.

Comment #3:

Also, recent literature (scientific report, ACS Nano) on the energy aspect, and also the characterization (chemical for instance) of the production of graphene from biomass is missing in this paper. This could have been useful to understand the phase partitioning (char, bio-oil and gas). In fact, based on the literature in the field, the graphene is not directly produced unless the unless a catalyst (Ca, Fe, Ni,etc...) is used. What is obtained is a biochar with a certain graphitization degree as shown in the supplementary materials in Figures 6 to 9 (Raman Spectra, XPS). Based on this, claiming on the title that graphene os made out of this process is not correct. Unstead, the authors could be right to claim a certain graphitization degree (with a certain amount of graphene sheets in the biochar)
The supplementary materials are not providing this information either.

So, despite the quality of experimental methods and interpretation of results that is provided, this work does not have the depth of analysis or the generic value that would justify publication.

The authors failed in checking the state-of-the-art in the field that could have been useful to better put together their paper.

Based on this assessment, I recommend rejecting this paper

Reviewer #3 (Remarks to the Author):

Comments:

The authors continuously fabricated graphene using renewable biomass material with integrated devices. Meanwhile, their production method was applicable to various biomass materials, suggesting the high-value application potential in a wide range of fields. Importantly, the production rate of the fabrication process was impressively high (28 s batch-1). The finding is quite interesting and strong enough. This work would make great contribution to the large-scale synthesis of graphene from renewable biomass. The magnitude of this work and the coherence of the paper structure make it highly suitable for publication in a prestigious journal. Overall, I think this manuscript is acceptable for publication in Nature Communications after the following issues are addressed:

1. The pyrolysis pre-treatment is one of the key novel points in this paper. Why is this step crucial to achieving low-carbon production?
2. It is mentioned in the manuscript that biochar with various pyrolytic temperature was used as raw materials to produce graphene. Can all these raw materials produce high-quality graphene?
3. Is there any difference between the graphene produced in this paper and the graphene produced by traditional technology?
4. This paper describes a very high production rate of graphene. Is there any technology to increase yield further in the future?

Reviewer #1 (Comments for the Author):

Comments: I am satisfied that the authors have responded to my comments and recommend this submission for publication, provided that the authors also include in the manuscript the information below that they included in their responses to me. The rebuttal letter should not merely be a communication with the review, but this needs to also be transmitted in the manuscript, please. After the small additions noted below, it will be ready. I do not need to review it again.

Q1. At the top of page 7, the authors state, “And it is higher than the first and second-generation fabricating technology.” Which fabrication technology are the authors referencing? They already mentioned flash graphene fabrication technology in the previous sentence.

The authors should also use the word “fabrication” instead of “fabricating”.

“Augmenting the capacitors” is also mentioned. What about the capacitors should be augmented? The number, the voltage, or the capacitance?

A: (1) First-generation fabrication technology means the biomass flash graphene by FJH at lab-scale (Refs. 1 and 5). The second-generation fabrication technology (Ref. 4) exhibits a high production rate at the expense of the pumping step1. This has already explained in the legend in Figure 1.

(2) As your suggestion, we have changed the word “fabricating technology” to “fabrication technology”.

(3) “Augmenting the capacitors” refers to increase the number of series capacitors, thereby increasing the discharge voltage and ultimately improving the sample reaction mass of a single batch.

***Please clarify in the manuscript that you are referring to the number of capacitors*

in series. **

Reply: Thank you for the good comment for this work. The capacitors used in lab-scale flash graphene production device is 63.8 mF, while in pilot-scale fabrication device is 127.6 mF. Therefore, we have added this information in Supplementary Table 2.

Q2. Figure 3. Why is exfoliation required for flash graphene production? Exfoliation is typically used for top-down methods, but not for bottom-up methods like flash Joule heating. The main text leads to extended data figures 5a-5b for more information on inadequate exfoliation, but the figures themselves do not discuss this.

A: We found that insufficient graphitization leads to incomplete graphene structure conversion. In extended data figures 5a-5b, the low sample-to-device resistance ratio (2.4) of high-temperature biochar-based preliminary FG led to insufficient voltage and temperature allocation at relatively low DC discharge voltage, resulting in inadequate graphitization.

Please include this explanation somewhere in the manuscript.

Reply: As your suggestion, we have added the following explanation of flash graphene fabrication mechanism in lines 154-155 of the manuscript. “However, the low sample-to-device resistance ratio (2.4) of high-temperature biochar-based preliminary FG led to insufficient voltage allocation at relatively low DC discharge voltage, resulting in inadequate graphitization.”

Q3. Supplementary table 5, 10, 11, 13, 14: how is H/C calculated? It does not appear to be H%/C% You should also be specific about whether your quantities are in wt% or atomic %.

A: The hydrogen-carbon ratio is calculated as follows: $H/C = (H \text{ relative weight} / H \text{ molecular weight}) / (C \text{ relative weight} / C \text{ molecular weight})$. H relative weight and C relative weight used in this equation are percentage by weight.

Please include this information under the corresponding supplementary figures.

Reply: Thank you for your suggestion. According to the equation of H/C , we added the detailed explanation of H/C in Supplementary table 5, 6, 10, 11, 13, 14.

Reviewer #2 (Comments for the Author):

Comments: Comments on the paper NCOMMS-23-60727-T "Continuous and low-carbon production of biomass flash graphene" by Zhu et al.

The paper addresses an important Topic consisting in making graphenic-like materials from bioresources.

Reply: In this study, we have achieved continuous biomass flash graphene production by using *state-of-the-art* flash Joule heating (FJH) technology *rather than pyrolysis*. The emerging FJH technology has been developed over the past three years to produce high-quality flash graphene from biomass (Nature, 2020, 577, 647-651)¹. FJH can generate a *current that directly through the carbon precursor*, producing an instantaneous (second level) high temperature for graphitization and simultaneously causing the graphite sheet to be exfoliated off by the current.

However, as you said, traditional *pyrolysis can only produce graphite-like material* by thermal radiation due to the lack of exfoliation process. The usage of “*pyrolysis-FJH nexus*” in this paper may have caused a misunderstanding for the reviewer that we employ pyrolysis for graphene synthesis. The detailed flash graphene production process is as follows. Biomass is first pyrolyzed (300-900 °C, 2h) to release pyrolytic volatiles and obtain biochar for further flash graphene fabrication by FJH. And then, biochar is further treated *by alternating current flash Joule heating (AC-FJH)* to produce preliminary flash graphene *via* electric-induced high temperature (~ 2000 K, 6 s). Finally, preliminary FG is exfoliated *by alternating current flash Joule heating (DC-FJH)* to form flash graphene *via* ultrahigh current (~150 A, 30 ms). DC-FJH reaction can produce a higher reaction temperature than AC-FJH for further

graphitization, and through the instantaneous ultrahigh current (~ 150 A) to achieve exfoliation of graphite sheet.

Current research on FJH focuses on lab-scale device creation, FJH reaction mechanism, and flash graphene application¹⁻⁸. However, there are still two drawbacks that urgently need to be addressed: (1) lacking integrated device; (2) the production of flash graphene with high carbon emission due to the energy waste on the biomass pyrolytic volatiles release, rather than structure optimization of graphene during AC-FJH.

To fill these gaps, (1) we created an integrated automatic device to achieve continuous biomass flash graphene production. (2) Fabrication mechanism of flash graphene includes carbonization graphitization and exfoliation, but the high temperature (~ 2000 K) employed by AC-FJH for volatiles release (carbonization) is energy-wasting. Obviously, volatiles release can be performed at lower temperatures ($\sim 500^\circ\text{C}$) and does not require such a high-temperature AC-FJH process. Meanwhile, pyrolysis is considered as an energy self-sufficient technology. Hence, pyrolysis was used to remove the biomass pyrolytic volatiles before FJH reaction. And the obtained solid phase product (biochar) was used as carbon precursor for further graphene production by FJH. Therefore, we proposed *pyrolysis-FJH nexus strategy* with reasonable energy allocation to achieve low-carbon production of flash graphene. Furthermore, we verify the low carbon emission of flash graphene production path through *material flow* and *life cycle assessment*.

Specific comments:

Comment#1:

In Figure 2b, the biogas formation is claimed which is impossible because of the low Hydrogen content that is removed in the volatile at the drying and/ or pyrolysis step. The same comment is valid for Fig3a and 3b where the amount of volatiles is questionable. In fact, Bamboo used as biomass source is known to have very high silica content (inorganic composition not provided in this paper) that inhibit the pyrolysis and favor the biochar formation rather than the bio-oil or gas production. This is widely documented in the literature and is in contradiction with the author claims that are not substantiated. The authors failed in demonstrating an appropriate understanding of the pyrolysis of biomass.

Reply: We thank the reviewer for this question. In Figure 2b in the manuscript, the “pyrolysis” is written above the “biochar” in life cycle assessment (LCA) system boundary figure, which may make a misunderstand that we re-pyrolyzing the biochar. In fact, we pyrolyzed biomass to release volatiles and produce biochar. And then, biochar is treated as carbon precursors for further flash graphene production by FJH. Therefore, we have modified LCA system boundary diagram to put each production process (pyrolysis, AC-FJH, and DC-FJH) directly above the carbon precursor (Figure 1). In addition, we have tested that the ash content of sawdust and bamboo powder used in this paper is less than 1%, so there may be no inhibition effect of pyrolysis as mentioned by reviewer.

Original figure in Figure 2b in the manuscript

Blue circle: The “pyrolysis” is written above the “biochar” in system boundary diagram, which may make a misunderstand that we re-pyrolyzing the biochar.

Modified figure

Figure 1. The figure above shows the system boundary diagram shown in the original manuscript. The figure below shows the modified figure. *note: “Loss1”

refers to the bio-oil and depletion in AC-FJH. #note: “Loss2” refers to the pyrolytic volatiles (probably bio-oil or gas) and depletion in DC-FJH. @note: In the AC-FJH reaction, bio-gas is produced only in the path of biomass and 300°C biochar as feedstock.

Comment #2:

In Figure 3a: the pyrolytic volatiles rate seems not realistic so are the data on Figure 3b if we refer the composition of the initial feedstock and to the literature. Consequently, the data on Figure 3c are questionable too. For this figure, the authors could have substantiated the data and claims in providing the proximate and ultimate analysis and see the H/C and O/C ratio for instance.

Reply: In this study, we proposed a pyrolysis-FJH nexus to achieve continuous biomass flash graphene production with low carbon emissions. The detailed production process involves three stages: Pyrolysis is used to release volatiles and obtain biochar; AC-FJH is utilized for high-temperature graphitization of biochar, achieving graphite-like materials (preliminary flash graphene); DC-FJH is employed for high-current exfoliation of preliminary flash graphene to produce flash graphene.

In pyrolysis process, the yield of 300°C biochar is 41.3%, the yield of 600°C biochar is 22.4%, the yield of 750°C biochar is 21.0%, the yield of 900°C biochar is 19.3% in Paths B-E (Table 1 and Table 2). In addition to biochar, the yield of bio-oil and bio-gas is 58.7%, 77.7%, 79.0%, and 80.7% at the pyrolysis temperature of 300-900 °C. *These phase partitioning are consistent with the reported literature*^{9, 10}.

Table 1. Phase partitioning (pyrolysis, AC-FJH, DC-FJH) of flash graphene (FG) from 10 gram sawdust.

	Pyrolysis		AC-FJH				DC-FJH				
	Biochar (g)	Bio-oil & bio-gas (g)	Bio-gas (g)	Preliminary FG (g)	Loss1 (g)	Temperature (K)	Current (A)	FG (g)	Loss2 (g)	Temperature (K)	Current (A)
Path A	-	-	5.17	1.34	4.01	1870	20.0	1.11	0.24	2600	68
Path B	4.13	5.87	1.00	2.29	0.34	1490	21.6	1.94	0.44	2830	96
Path C	2.24	7.77	-	2.10	0.25	1330	23.6	1.87	0.03	2680	144
Path D	2.10	7.90	-	1.93	0.17	1430	22.7	1.73	0.02	2830	164
Path E	1.93	8.07	-	1.69	0.24	1230	24.0	1.61	0.01	2400	152

Path A: Biomass is first treated by AC-FJH to produce preliminary FG, and then preliminary FG is fabricated by DC-FJH to produce FG;

Path B: Biomass is first pyrolyzed to release pyrolytic volatiles and produce *300°C biochar*, then biochar treated by AC-FJH to produce preliminary FG, and finally preliminary FG is fabricated by DC-FJH to produce FG;

Path C: Biomass is first pyrolyzed to release pyrolytic volatiles and produce *600°C biochar*, then biochar treated by AC-FJH to produce preliminary FG, and finally preliminary FG is fabricated by DC-FJH to produce FG;

Path D: Biomass is first pyrolyzed to release pyrolytic volatiles and produce *750°C biochar*, then biochar treated by AC-FJH to produce preliminary FG, and finally preliminary FG is fabricated by DC-FJH to produce FG;

Path E: Biomass is first pyrolyzed to release pyrolytic volatiles and produce *900°C biochar*, then biochar treated by AC-FJH to produce preliminary FG, and finally preliminary FG is fabricated by DC-FJH to produce FG.

Table 2. Phase partitioning of 1 gram flash graphene production process (pyrolysis, AC-FJH, DC-FJH) from various paths. The data in the table are converted from the data in Table 3 according to the yield.

	Pyrolysis (g)		AC-FJH (g)			DC-FJH (g)	
	Biochar	Bio-oil and bio-gas	Bio-gas	Preliminary FG	Loss1	FG	Loss2
Path A	-	-	4.67	1.21	3.61	1.00	0.21
Path B	2.29	3.26	1.00	1.29	0.21	1.00	0.29
Path C	1.26	4.37	-	1.13	0.13	1.00	0.13
Path D	1.21	4.55	-	1.11	0.10	1.00	0.11
Path E	1.20	5.00	-	1.05	0.15	1.00	0.05

In the AC-FJH process, high temperature (~2000 K) and high current (~25A) can be generated. According to the aromatization degree of carbon precursors (biomass or biochar), we divide the production pathways into two categories for discussion. Firstly, the yield of preliminary flash graphene from biomass and 300°C biochar is low (13.4% and 55.4%, Table 3). This is because biomass and 300°C biochar as carbon precursors have a low aromatization, as proved by the high H/C (Table 3). Thus, 51.7% and 24.9% of non-condensable bio-gas (including H₂, CH₄, CO, CO₂, C₂H₂, C₂H₄, C₂H₆, C₃H₆, C₃H₈) were quantitatively determined from AC-FJH reaction of biomass and 300°C biochar, respectively (Supplementary Table 3). However, the bio-oil can be condensed on quartz tubes due to the ultrafast cooling rate (Figure 2). And, this part is difficult to collect and quantitatively analyze.

Secondly, the yield of preliminary flash graphene obtained from 600-900 °C biochar is similar and high, ranging from 93.8% to 87.5%. The yield of preliminary flash graphene from 600-900°C biochar is higher than that from biomass and 300°C

biochar due to the high aromatization degree of 600 - 900°C biochar. Such a high temperature far exceed the previous pyrolysis reaction (300-900 °C) and will prompt graphitization degree of preliminary flash graphene This process will inevitably cause the release of non-condensable bio-gas and condensable bio-oil, as confirmed by the increased H/C (Paths B-E, Table 3). Meanwhile, it is inferred that only a very small number of pyrolytic volatiles (possibly bio-gas and bio-oil) are released according to the increased carbon content (Paths A-E, Table 3), which is consistent with the phase partitioning opinion of the reviewer #2.

In addition, the sample depletion was inevitably caused by high-temperature induced ejection and adhesion of the sample on the copper wire mesh and quartz tube. Therefore, we named “Loss1” for the bio-oil and depletion. The mass of “Loss1” is equal to the mass of biochar minus preliminary flash graphene and bio-gas ($M_{\text{Loss1}} = M_{\text{biochar}} - M_{\text{preliminary flash graphene}} - M_{\text{bio-gas}}$). The high “Loss1” value in Paths A-B validate the high bio-oil content from biomass and 300°C biochar.

Figure 2 Digital photograph of sample tube. The quartz tube remains condensable bio-oil that are difficult to collect after AC-FJH.

Table 3. Elemental composition of biochar from pyrolysis, preliminary FG from AC-FJH, and FG from DC-FJH.

	Biochar			Preliminary flash graphene			Flash graphene		
	C (%)	H (%)	H/C	C (%)	H (%)	H/C	C (%)	H (%)	H/C
Path A	*	*	*	82.8	0.05	0.007	87.7	1.09	0.149
Path B	66.6	3.52	0.635	79.4	1.31	0.198	92.6	0.19	0.025
Path C	86.1	1.37	0.190	95.0	0.19	0.024	95.4	0.10	0.013
Path D	82.1	0.43	0.063	92.7	0.07	0.009	97.5	0.06	0.008
Path E	90.5	0.44	0.059	86.0	0.06	0.008	92.8	0.02	0.003

*note is the element composition of biomass (sawdust). C: 46.3%, H:7.03%, H/C:1.823

In DC-FJH process, the ultrahigh temperature and current (~3000 K, ~150 A) can be generated. Such an ultrahigh temperature far exceed the previous AC-FJH reaction (~2000 K) and will further prompt graphitization degree of preliminary flash graphene, as confirmed by the increased H/C (Paths B-E, Table 3). This process will inevitably cause the release of non-condensable bio-gas and condensable bio-oil. This can be confirmed by the increased carbon content (Paths A-E, Table 3). However, the amount of bio-oil and bio-gas produced is too small to be collected for qualitative and quantitative analysis. In addition, the *sample depletion was inevitably caused* by high current (~150 A) induced ejection of the sample on the copper wire mesh. Therefore, we named “Loss2” for the bio-oil and depletion. The phase partitioning shown by the AC-FJH and DC-FJH process is consistent with the research of FJH technology founder (Figure 3)^{2,7}.

In conclusion, the phase partitioning of pyrolysis and FJH shown in the paper is realistic and logical, and is also consistent with the reported literature. Therefore, we sincerely hope that the modified manuscript is acceptable for publication.

[REDACTED]

Figure 3. Product distribution after FJH from polythene with various resistances (Figure 2e, Advanced materials, 35, 48, 2306763-2306773).

Comment #3:

Also, recent literature (scientific report, ACS Nano) on the energy aspect, and also the characterization (chemical for instance) of the production of graphene from biomass is missing in this paper. This could have been useful to understand the phase partitioning (char, bio-oil and gas). In fact, based on the literature in the field, the graphene is not directly produced unless the unless a catalyst (Ca, Fe, Ni, etc...) is used. What is obtained is a biochar with a certain graphitization degree as shown in the supplementary materials in Figures 6 to 9 (Raman Spectra, XPS). Based on this, claiming on the title that graphene os made out of this process is not correct. Unstead, the authors could be right to claim a certain graphitization degree (with a certain amount of graphene sheets in the biochar)

The supplementary materials are not providing this information either. So, despite the quality of experimental methods and interpretation of results that is provided, this work does not have the depth of analysis or the generic value that would justify publication. The authors failed in checking the state-of-the-art in the field that could have been useful to better put together their paper. Based on this assessment, I recommend rejecting this paper.

Reply: The FJH has been developed to produce high-quality flash graphene with catalyst-free and ultrafast reaction process (second-level)¹. As you said, even with the presence of catalysts, traditional pyrolysis can only transform biomass into graphite-like materials through thermal radiation without exfoliation.

However, the graphene synthesized in this study exhibits a distinct 2D peak, and can be considered as high-quality graphene based on its high $I_{2D/G}$ value (Figure 4).

Meanwhile, high C=C ratios in XPS and thin layer structure in TEM demonstrate that flash graphene fabricated in this study shows an excellent graphene structure. In contrast, conventional pyrolysis can only produce graphite-like material (Table 4 and Figure 5). This is because the graphite-like material produced by pyrolysis only has graphitization and lacks the exfoliation.

Whereas, the thick sheets of graphene in Path E may lead to a misconception regarding the quality of our produced flash graphene (Figure 4a). This is because of the low resistance of high-temperature biochar (900 °C). Based on the voltage division principle of series circuit, the low resistance of high-temperature biochar led to insufficient voltage allocation. This will further result in the low Joule heating. The resistance of low-temperature and medium-temperature biochar is high, resulting in a higher voltage allocation and Joule heating. Undoubtedly, the high Joule heating promote the formation of thin graphene layer. However, the high-temperature biochar based few-layered graphene can be fabricated by increasing the discharge voltage. And the increased discharge voltage can further increase the voltage allocated to the carbon precursor (Figure 4c).

The presence of thick graphene sheets in Path E may lead to a misconception regarding the quality of our produced flash graphene (Figure 4a). This is attributed to the low resistance exhibited by high-temperature biochar (900 °C). This results in inadequate voltage allocation based on the voltage division principle of a series circuit, subsequently leading to insufficient Joule heating. In contrast, low-temperature and medium-temperature biochar exhibit higher resistances, enabling greater voltage allocation and enhanced Joule heating. Undoubtedly, this elevated Joule heating facilitates the formation of thin graphene layers. However, it is feasible

to fabricate few-layered graphene using high-temperature biochar by increasing the discharge voltage. Furthermore, an increased discharge voltage can further augment the allocated voltage on carbon precursor (Figure 4c).

Figure 4. Raman, XPS, and TEM of flash graphene from various production paths under 150 V DC-FJH reaction.

[REDACTED]

Figure 5. The Raman spectra of biomass graphene produced in the literature from Scientific reports^{11, 12}, ACS Nano¹³, and ACS Appl. Mater. Interfaces¹⁴.

Table 4. Comparison of graphene produced by pyrolysis and flash Joule heating.

Year	Journal	Method	Reaction time	catalyst	Biomass	$I_{2D/G}$
2013	JMCA ¹⁵	900°C Pyrolysis	1h	ZnCl ₂	Coconut shell	0.37
2014	RSC Advances ¹⁶	450°C Pyrolysis	24h	FeCl ₃	Wood	0.4
2015	Scientific reports ¹¹	Hydrothermal and 850°C Pyrolysis	2h	KOH etching	Auricularia	-
2016	Journal of Electroanalytical Chemistry ¹⁷	2600°C Pyrolysis	5 min	KOH etching	Wheat straw	0.61
2016	ACS Nano ¹³	700°C Pyrolysis	4h	AgNO ₃	PEO–PPO–PEO	-
2017	Nanomaterials ¹⁸	900°C Pyrolysis	3h	-	Palm kernel shell	-
2019	Scientific reports ¹⁹	800°C Pyrolysis	2h	AgNO ₃	Onion peels	-
2019	Polymers ²⁰	1100°C Pyrolysis	1h	Fe(NO ₃) ₃	Lignin	-
2020	Scientific reports ¹²	Microwave and 500°C pyrolysis	3h	HNO ₃ etching	Spent tea	-
2020	AMI ¹⁴	1100°C Pyrolysis	2h	KOH etching	Almond shells	-
	This work	Flash Joule heating	6s	Catalyst-free	Sawdust	1.06

“-” indicates that $I_{2D/G}$ value is not provided in these literatures.

Path A: Biomass is first treated by AC-FJH to produce preliminary FG, and then preliminary FG is fabricated by DC-FJH to produce FG;

Path B: Biomass is first pyrolyzed to release pyrolytic volatiles and produce 300°C biochar, then biochar treated by AC-FJH to produce preliminary FG, and finally preliminary FG is fabricated by DC-FJH to produce FG;

Path C: Biomass is first pyrolyzed to release pyrolytic volatiles and produce 600°C biochar, then biochar treated by

AC-FJH to produce preliminary FG, and finally preliminary FG is fabricated by DC-FJH to produce FG;

Path D: Biomass is first pyrolyzed to release pyrolytic volatiles and produce 750°C biochar, then biochar treated by AC-FJH to produce preliminary FG, and finally preliminary FG is fabricated by DC-FJH to produce FG;

Path E: Biomass is first pyrolyzed to release pyrolytic volatiles and produce 900°C biochar, then biochar treated by AC-FJH to produce preliminary FG, and finally preliminary FG is fabricated by DC-FJH to produce FG.

Reviewer #3 (Comments for the Author):

Comments: *The authors continuously fabricated graphene using renewable biomass material with integrated devices. Meanwhile, their production method was applicable to various biomass materials, suggesting the high-value application potential in a wide range of fields. Importantly, the production rate of the fabrication process was impressively high (28 s batch-1). The finding is quite interesting and strong enough. This work would make great contribution to the large-scale synthesis of graphene from renewable biomass. The magnitude of this work and the coherence of the paper structure make it highly suitable for publication in a prestigious journal. Overall, I think this manuscript is acceptable for publication in Nature Communications after the following issues are addressed:*

1. The pyrolysis pre-treatment is one of the key novel points in this paper. Why is this step crucial to achieving low-carbon production?

Reply: In the biochar-involved FG production path, we used low-energy pyrolysis to prioritize removing the volatiles released in the energy-intensive AC-FJH process, while AC-FJH was only used for further graphitization. Therefore, low-carbon biomass FG production can be achieved through appropriate energy allocation, which is called energy cascade requirement.

2. It is mentioned in the manuscript that biochar with various pyrolytic temperature was used as raw materials to produce graphene. Can all these raw materials produce high-quality graphene?

Reply: The flash graphene structure can be adjusted by increasing the sample-allocated voltage. It was noted that the few-layer structure could be achieved from

low- and medium-temperature biochar-based preliminary flash graphene due to appropriate sample-allocated voltage. However, the low sample-to-device resistance ratio of high-temperature biochar-based preliminary flash graphene led to insufficient voltage allocation at relatively low DC discharge voltage, resulting in inadequate graphitization. Therefore, the sample-allocated energy could be increased by increasing the voltage to achieve few-layered flash graphene derived from high-temperature biochar.

3. Is there any difference between the graphene produced in this paper and the graphene produced by traditional technology?

Reply: In terms of graphene structure (layers) and applications (dispersibility, catalytic performance, photothermal conversion, thermal conductivity), the pyrolysis-FJH nexus production technology of high-purity flash graphene outperforms the traditional production technology. However, the FJH in producing graphene is more efficient (seconds level) and without a catalyst.

4. This paper describes a very high production rate of graphene. Is there any technology to increase yield further in the future?

Reply: We can further increase biomass FG production by increasing the number of capacitors and sample loading weight with next generation flash graphene production device. It can run multiple devices at the same time. Development of the next generation of products is already under way.

Reference

- (1) Luong, D. X., et al., Gram-scale bottom-up flash graphene synthesis. *Nature* **2020**, 577, 647-651.
- (2) Deng, B., et al., Urban mining by flash Joule heating. *Nat. Commun.* **2021**, 12, 5794-5801.
- (3) Deng, B., et al., Phase controlled synthesis of transition metal carbide nanocrystals by ultrafast flash Joule heating. *Nat. Commun.* **2022**, 13, 262.
- (4) Deng, B., et al., Urban mining by flash Joule heating. *Nat. Commun.* **2021**, 12, 5794.
- (5) Wyss, K. M., et al., Large-scale syntheses of 2D materials: Flash Joule heating and other methods. *Adv. Mater.* **2022**, 34, 2106970.
- (6) Beckham, J. L., et al., Machine learning guided synthesis of flash graphene. *Adv. Mater.* **2022**, 34, 2106506.
- (7) Wyss, K. M., et al., Synthesis of clean hydrogen gas from waste plastic at zero net cost. *Adv. Mater.* **2023**, 35, 2306763.
- (8) Advincula, P. A., et al., Flash graphene from rubber waste. *Carbon* **2021**, 178, 649-656.
- (9) Mishra, R. K.; Mohanty, K., Pyrolysis kinetics and thermal behavior of waste sawdust biomass using thermogravimetric analysis. *Bioresour. Technol.* **2018**, 251, 63-74.
- (10) Guo, Y.; Wang, Q., Exploring the adsorption potential of Na₂SiO₃-activated porous carbon materials from waste bamboo biomass for ciprofloxacin rapid removal in wastewater. *Environmental Technology & Innovation* **2023**, 32, 103318.
- (11) Zhu, Z., et al., Dual tuning of biomass-derived hierarchical carbon nanostructures for supercapacitors: The role of balanced meso/microporosity and graphene. *Sci. Rep.* **2015**, 5, 15936.
- (12) Abbas, A., et al., High yield synthesis of graphene quantum dots from biomass waste as a highly selective probe for Fe³⁺ sensing. *Sci. Rep.* **2020**, 10, 21262.
- (13) Jiang, H., et al., Self-volatilization approach to mesoporous carbon nanotube/silver nanoparticle hybrids: The role of silver in boosting Li ion storage. *ACS Nano* **2016**, 10, 1648-1654.
- (14) Pham, H. D., et al., Dual carbon potassium-Ion capacitors: biomass-derived graphene-like carbon nanosheet cathodes. *ACS Appl. Mater. Interfaces* **2020**, 12, 48518-48525.
- (15) Sun, L., et al., From coconut shell to porous graphene-like nanosheets for high-power supercapacitors. *J. Mater.*

Chem. A **2013**, *1*, 6462-6470.

(16) Akhavan, O., et al., Synthesis of graphene from natural and industrial carbonaceous wastes. *RSC Advances* **2014**, *4*, 20441-20448.

(17) Chen, F., et al., Facile synthesis of few-layer graphene from biomass waste and its application in lithium ion batteries. *J. Electroanal. Chem.* **2016**, *768*, 18-26.

(18) Nasir, S., et al., Oil palm waste-based precursors as a renewable and economical carbon sources for the preparation of reduced graphene oxide from graphene oxide. *Nanomaterials* **2017**, *7*, 182.

(19) Bhujel, R., et al., Capacitive and sensing responses of biomass derived silver decorated graphene. *Sci. Rep.* **2019**, *9*, 19725.

(20) Li, J., et al., Efficient conversion of lignin waste to high value bio-graphene oxide nanomaterials. *Polymers* **2019**, *11*, 623.

REVIEWERS' COMMENTS

Reviewer #2 (Remarks to the Author):

The authors have correctly addressed the comments. I recommend this paper for publication

Reviewer #3 (Remarks to the Author):

Accepted.